# Discrepancy between Food Classification Systems: Evaluation of Nutri-Score, NOVA Classification and Chilean Front-of-Package Food Warning Labels

**DOI:** 10.3390/ijerph192214631

**Published:** 2022-11-08

**Authors:** Aranza Valenzuela, Leandro Zambrano, Rocío Velásquez, Catalina Groff, Tania Apablaza, Cecilia Riffo, Sandra Moldenhauer, Pamela Brisso, Marcell Leonario-Rodriguez

**Affiliations:** 1Facultad de Medicina y Ciencias de la Salud, Escuela de Nutrición y Dietética, Universidad Mayor, Temuco 4810767, Chile; 2Centro de Biología Molecular y Farmacogenética, Departamento de Ciencias Básicas, Facultad de Medicina, Universidad de La Frontera, Temuco 4810767, Chile

**Keywords:** ultra-processed foods, open food facts, food labelling

## Abstract

Background: Currently, there are different food classification systems in order to inform the population of the best alternatives for consumption, considering all the diseases associated with the consumption of products of low nutritional quality. Reports indicate that these forms of labelling warnings correspond to a laudable strategy for populations that do not have the knowledge to discriminate between the wide range of products offered by the food industry. However, recent publications indicate that there may be inconsistencies between the different classification guidelines, and the guidelines that nations should adopt in their food guides are still a matter of debate. In view of this, the present study aimed to evaluate the quantitative and qualitative differences that exist between the NOVA, Nutri-Score and Chilean Front-of-package (FoP) food warning label according to the Chilean basic food basket list. Method: An analytical study was carried out to classify a list of 736 foods according to three different systems, evaluating the distributions according to their methods of classifying the products. Quantitative differences were contrasted for each system, as well as between them, together with an analysis of the dimensions of each system. Results: According to the Nutri-Score classification, the most frequent category was A with 27% (high nutritional quality), followed by D with 22% (low nutritional quality) of the total. On the other hand, the NOVA classification showed that the most frequent categorization was ultra-processed food (NOVA 4) with 54%, followed by unprocessed (NOVA 1) with 19%. Regarding the FoP warning labels, 57% of the foods were categorized as free warning labels, followed by the category of foods with 3 warning labels (23%). Regarding the results of the principal component analysis, the Nutri-Score and FoP warning labels present a degree of similarity in their classification guidelines, being different than the dimension pointed out by NOVA. Conclusion: The present work managed to demonstrate that there are quantitative and qualitative differences between the classification and recommendation guidelines of the Nutri-Score, NOVA and FoP warning labels, finding concrete discrepancies between them.

## 1. Introduction

The prevalence of non-communicable diseases (NCDs), including cardiovascular diseases, cancer, diabetes, and chronic lung diseases, has increased in all age groups and is a major cause of disability and premature death in Latin America and the Caribbean [1,2]. One of the main factors influencing this situation is the excessive consumption of ultra-processed products, for which there is significant evidence linking their intake to the onset of various diseases [3] and increased all-cause mortality [4,5,6]. These products are characterized by a high calorie content and poor nutritional value, established by their high levels of sugar, fat and sodium, as well as being deficient in essential micronutrients [7]. Moreover, in different parts of the world, they constitute a cheaper and more accessible dietary alternative for the population [8], creating a complex dilemma for economically vulnerable communities, since, due to their limited food education, they often opt for the high consumption of this group of foods [9,10].

This problem is one of the main health challenges worldwide, and the development of policies and actions that allow the population to cope with this situation is a priority for various governments. In this line, food classification systems have emerged, which allow for the identification of the characteristics of the products to be consumed, either by indicators, phrases or standardized elements previously established by the health authorities of each country [11]. In Chile, front-of-package (FoP) food warning labels were implemented by law Nº 20.606 on the nutritional composition of food and its advertising, in order to inform the population about products with an “excess” of certain nutrients critically responsible for obesity and cardio-metabolic diseases, with positive results reported since its implementation [12,13]. However, this labelling law does not consider the degree of processing of the food, as integrated in the NOVA classification system proposed in 2010 by researchers in Brazil [14]. On the other hand, warning labels focus on the negative elements of the food composition without considering the presence of positive components for the health of the consumer, as is the orientation of the Nutri-Score rating system proposed by French researchers a decade ago [15]. In the face of this offer of rating guidelines, an understanding of how these tools behave in classifying foods from each country would allow for complementing their guidelines to refine actions to better educate consumers. However, recent publications report significant discrepancies between Nutri-Score and NOVA recommendations [16]. In addition to this, research that considers Chile’s FoP warning labels is limited to other types of research [12,17], with no reports to inform health authorities about which guidelines are more positive for Chilean consumers to select food in a correct, responsible, healthy, and informed way. Against this background, the present work aims to evaluate the quantitative and qualitative differences that exist between the NOVA, Nutri-Score and FoP warning labels according to the list of the basic food basket in Chile.

## 2. Materials and Methods

An analytical study was carried out based on the evaluation of a list of foods belonging to the Chilean basic food basket, which were available in the online catalogue of a supermarket widely accessible to the Chilean population. It is important to mention that the Chilean basic food basket is a list of 80 products, which are monitored monthly by the Ministry of Social Development of the Chilean Government with figures from the National Institute of Statistics, to review the consumption habits of Chilean households and determine the poverty line.

Regarding the selected establishment, it derives from a transnational corporation that is found in all the regional capitals of the country and corresponds to one of the three most important hypermarkets in Chile, characterized by its variety of products and economic prices. The selection of foods from the catalogue is specifically spelled out in Figure 1.

### 2.1. Ranking Systems

#### 2.1.1. Nutri-Score

The classification system proposed by the group of French scientists led by Serge Hercberg proposes a nutritional traffic light that establishes a front-end labelling based on nutritional quality according to five different letters (A, B, C, D and E), associated with a specific colour for each one (Figure 2a). The classification formula is based on a continuous quantitative indicator ranging from −15 points (healthiest) to +40 (least healthy). Energy, sugar, fat and sodium content score points that contribute to a negative ranking, while the presence of fruits, vegetables, nuts, legumes, protein and fibre contribute to a positive ranking. According to the total sum of each of the scores, the food is classified as A when the score is between −15 and −1, B when the addition is between 0 and 2, C between 3 and 10, D for scores 11 and 18, and E when the value is between 19 and 40 points. The A/B classification is for recommended foods, while the letters D/E classify products of poor nutritional quality. Ranking was performed independently by two researchers of the group using the official NutriScore Calculation Tool software of Santé Publique France^®^ (https://www.santepubliquefrance.fr/). The typed ratings were then compared separately to check for possible typing or data processing errors.

#### 2.1.2. NOVA System

The NOVA system was proposed by the NUPENS research group at the Faculty of Public Health of the University of Sao Paulo in Brazil, led by researcher Dr. Carlos Monteiro. This classification establishes that foods can be categorized according to their degree of processing, proposing that NOVA 1 corresponds to unprocessed or minimally processed foods, NOVA 2 to processed culinary ingredients, NOVA 3 to processed foods, and finally NOVA 4 to ultra-processed products (Figure 2b). This last group is characterized by products with a poor food matrix, concentrated in additives and refined ingredients such as coloring agents, additives, thickeners, emulsifiers, etc. As with the other systems, two researchers independently classified all the foods on the list individually. Subsequently, typing results were compared to resolve discrepancies.

#### 2.1.3. FoP Warning Labels

Chile’s system of warning labels is a result of the Labelling Law 20.606, which came into force in its final stage in 2019. It regulates those foods that have added sugars, saturated fat or sodium, exceed the limits established by the Chilean Ministry of Health for these nutrients and/or calories and therefore must display a warning stamp indicating this on their front side (Figure 2c). For solids, the corresponding limit values for each element are energy (275 kcal/100 g), sodium (400 mg/100 g), sugars (10 g/100 g) and saturated fats (4 g/100 g). For liquids, the values are as follows: energy (70 kcal/100 mL), sodium (100 mg/100 mL), sugars (5 g/100 mL) and saturated fat (3 g/100 mL). For evaluation of the present system, two independent researchers classified each listed food according to the nutritional composition per 100 g/100 mL of product as: without warning labels, and one, two, three and four warning labels. Following the classification, double digit results were compared separately to assess discrepancies and resolve them on an individual basis.

#### 2.1.4. Data Analysis

Microsoft Excel^®^ (Microsoft, Redmond, WA, USA) spreadsheets were used for data processing, where each researcher in charge of classifying the foods presented their results independently. At the end of this stage, and after review by the responsible researcher, the final database with all the foods and their respective classifications was created. For this purpose, it was exported to GraphPad Prism^®^ v9.0 (San Diego, CA, USA) software to determine frequencies by food groups (cereals, potatoes and fresh pulses as group 1, dairy products as group 2, fish, meat, eggs and dried pulses as group 3, oils and fats as group 4, and finally sugars as group 5. Furthermore, to establish a comparison between the quantitative discrepancies between the different systems, comparative analysis was applied through cross-analysis. Finally, to understand the different qualitative orientations of each system, principal component analysis was performed for each system and food group.

## 3. Results

Of all the foods included in this study (*n* = 736), according to the Nutri-Score classification, the most frequent category was A with 27%, followed by D with 22% of the total. On the other hand, the NOVA classification showed that the most frequent categorization was ultra-processed food (NOVA 4) with 54%, followed by unprocessed (NOVA 1) with 19%. About the FoP warning labels, 57% of the foods were categorized as warning label-free, followed by the category of foods with 3 warning labels (23%). When analyzing these results as a whole, we can see that for the NOVA system most of the foods on the list (*n* = 394) would not be considered as recommended, while for the other two systems, the reality is the opposite, placing most of the foods analyzed in their recommended classification (Nutri Score A—*n* = 197; No warning labels—*n* = 421) (Figure 3). Regarding the distribution of the different categories by each classification system, Nutri-Score presents more homogeneous results for each of its groups, contrasting sharply with the heterogeneity established by NOVA and FoP warning labels, where majority groups are evidenced by over 50%.

Regarding the results by food groups, for cereals (*n* = 198), according to Nutri-Score, the category with the highest frequency was again A with 46%, followed by category E with 19%. Regarding NOVA, ultra-processed foods (NOVA 4) were in first place (65%), followed by unprocessed foods (NOVA 1) (25%). Finally, according to the system of FoP warning labels, it was found that 49% corresponded to the category of food without warning labels (0 seals) and 29% corresponded to food with 3 warning labels (Figure 3). As in the global evaluation of the list, in cereals there is a similar distribution with respect to the heterogeneity established by the NOVA and FoP warning labels categories; however, in this group of foods Nutri-Score is added with the predominance of its category A (46%). Again in cereals, the trend is incongruent between the systems according to the consumption recommendations of their guidelines.

For the dairy group, a total of 109 products were considered, of which according to the Nutri-Score classification, the predominant category was B with 47%, followed by D with 19%. For the NOVA classification, the category with the highest frequency was ultra-processed foods (NOVA 4), followed by NOVA 3, corresponding to processed foods. Finally, according to the FoP warning labels, it was observed that 75% of dairy products correspond to the classification of foods without warning labels and 18% correspond to foods with 1 warning label (Figure 3). As with the cereals group, the contradictory trend regarding the recommendation of each system is maintained.

For the group of meat, eggs, fish and pulses (*n* = 139), 83 foods (60%) were classified in group A according to the Nutri-Score. For the NOVA system, 67 foods (48%) were classified as unprocessed foods (NOVA 1). Finally, for the FoP warning labels, 99 foods (71%) were classified as without warning labels (Figure 3). Unlike the two previous food groups, here there is congruence between the food products that are considered recommended for the population.

Regarding oils and fats, 100 products were considered, 51% of which correspond to group C of the Nutri-Score system, 64% to processed culinary foods (NOVA 2) and 50% to foods free of warning labels (Figure 3). Finally, for the sugars and others group (*n* = 190), according to the Nutri-Score classification, E predominates with 33%, followed by group D with 25%. For the NOVA classification, the category with the highest frequency was ultra-processed food (NOVA 4) with 75%, followed by culinary processed food (NOVA 2) with 11%. Regarding the FoP warning labels, the most prevalent was free warning labels food with 49% (Figure 3). In this food group was the only instance where NOVA and Nutri-Score were mostly categorized with their non-recommended ratings.

To contrast the quantitative differences that were evident in each classification separately, a cross analysis was carried out between all the systems according to the global list of foods. In the first instance, Nutri-Score was contrasted with the NOVA classification (Table 1), showing similarities and inconsistencies according to the categorization of each system. In the first instance, of all the A products, 53.4% correspond to NOVA 1; however, another figure (18.7%) would be categorized as an ultra-processed product, which is not recommended according to the NOVA classification. Along the same lines, 70.3% of all B products correspond to NOVA 4, making the incongruence between the two systems even more concrete in terms of their recommendations. It is important to note that when categorizing the products least recommended by each system (E and NOVA 4), there is 88.1% compatibility.

Regarding the comparison between Nutri-Score and FoP warning labels (Table 2), a great congruence is evident between the most recommended categories of each system (A and Without Warning Labels) of 94.9%. Regarding NutriScore category B, most of the classified foods are concentrated without seals (85.5%) or at least one (10.1%). Regarding foods considered as not recommended, for category D there is a homogeneous distribution according to the classification by warning labels, the differences being more concrete in distribution for E products, where most of them have three seals (77.1%). While there are similarities in the classification of recommended products rather than NOVA, this phenomenon is not observed for the non-recommended categories.

The contrast between NOVA and FoP warning labels (Table 3) shows a total congruence between products classified as NOVA 1 and absence of warning labels (100%). In the same line, the majority pattern is repeated for NOVA 2 (77.5%) and NOVA 3 (71.5%) the absence of warning labels. However, worryingly, for the NOVA 4 category there are a large number of products without warning labels (33.2%), being the second highest frequency after three warning labels (38.9%).

Finally, and to better understand the quantitative differences expressed in the results presented above, a principal component analysis was carried out to show which dimensions are targeted by each system, evaluating the overall list, as well as by food group (Figure 4). Regarding all products, the Nutri-Score and FoP warning labels present a similarity in their classification guidelines, being different than the dimension targeted by NOVA. Regarding cereals, this trend is repeated, and even a greater closeness between the Chilean system and Nutri-Score is exposed, as well as the trend that NOVA and Nutri-Score are further apart. Dairy products are the food group where there are the greatest differences between the three systems, and even the dimension to which NOVA points is totally contrary to Nutri-Score, and FoP warning labels is maintained as an intermediate. The latter is maintained in all subsequent groups, along with a greater or lesser distance between the systems that presented a greater discordance (NOVA and Nutri-Score).

## 4. Discussion

The increase in the intake of ultra-processed foods has been evidenced in different population groups around the world, reporting a significant increase in the energy consumed from these products by non-Hispanic black youth (from 62.2% to 72.5%; [95% CI, 6.8% to 13.8%]), Mexican Americans (from 55.8% to 63.5%; [95% CI, 4.4% to 10.9%]) and non-Hispanic whites (from 63.4% to 68.6%; [95% CI, 2.1% to 8.3%]) over the last 10 years [18]. This situation is associated with impoverished diet quality [19], overnutrition [20] and increased all-cause mortality [21], with people with lower educational and socio-economic status being the most affected [9,10]. Understanding that the gaps in nutritional education and the food industry’s marketing deepen this problem [8], the use of food classification systems by health organizations would help the population to make better choices about what they consume. Along these lines, in the last decade, different classification systems have emerged, and from this, proposals for consumption recommendations, with inconsistencies between them [16]. It is therefore necessary to analyse and study consumption recommendations before renewing public policies in this field. In this sense, the research here is a necessary input for the discussion of public health and nutrition decisions to be taken over the coming years.

Our results indicate differences would be established mainly by the principles considered by each system; warning labels base their verdict on a cut-off point for calories, sugars, saturated fat and sodium [17]; Nutri-Score includes these parameters in a continuous and quantitative way and includes in the assessment the positive characteristics of the products, such as protein content, fiber content and the inclusion of plant foods [15]. With respect to the NOVA classification, it focuses on characteristics regarding the nutritional quality according to the degree of processing of the product [22], aiming at a totally different dimension from the two previous systems. This would explain the more marked discriminatory nature of the results obtained, since in all the food groups, as well as in the overall list, the predominant classification was NOVA 4. This could be explained by the origin and nature of the list of products analyzed, since it is known that most of the supply in supermarkets tends to concentrate highly processed products, as reported by Hernández et al. [23]. In this line, regardless of whether the product has positive characteristics such as added fiber or protein, or even no seal at all, its level of processing would not allow it to be in NOVA 1, 2 or 3. In contrast, for the other systems, greater homogeneity is observed, possibly explained by the ability of some products to move between different classifications despite belonging to the same group. This is the case for some milk drinks which, when added with proteins or fiber, can obtain positive categories such as A or B even if they contain sugars or sodium. This is also true for the FoP warning labels, since, by replacing sugar with non-caloric sweeteners, they can avoid the high-calorie or high-sugar labels. This situation has been reported by some authors who postulate that these forms of classification would allow for the whitening of ultra-processed products, a capacity that is exploited by the food industry, and a strategy that is not possible to shape under NOVA guidelines and the degree of food processing, resulting in strong debate in this area [24].

Conflict could imply shortcomings regarding the consideration of a product as recommended or not; however, as a tool to discriminate between the nutritional poverty of foods it would be quite adequate [25], especially with Nutri-Score that acts under continuous and not nominal quantitative parameters. These guidelines allow for greater sensitivity to discriminate according to the content of critical nutrients, making them more useful in food environments where the options are not varied, or that must necessarily appeal to the consumption of ultra-processed foods, such as in food swamps [26].

This mathematical objectivity that NOVA does not present generates a lot of criticism in certain academic sectors, as there are reports that its parameters are not even recognized by food and nutrition specialists [27], making it less useful in practice for the general population. However, it is important to mention that the detected shortcoming could be remedied by machine learning models, some of which have already been developed (FoodProX Algorithm), and which clearly offer an opportunity to reduce the subjectivity established by the qualitative parameters of NOVA [28]. These computer algorithms are based on the distribution of macronutrients and micronutrients in unprocessed foods versus their counterparts, which have derisory distributions in 100 g or mL of product, content that could never be reported in foods of NOVA categories 1, 2 or 3, for example.

In terms of cross-system results, the present study is the first effort to evaluate the differences between the three systems considered. Previously, Ferreiro et al. [16] contrasted NOVA and NutriScore, to which we have added to the analysis the warning labels proposed by the Ministry of Health of the Government of Chile in 2016. Regarding data reported in the 2021 publication, the overall distribution is quite consistent, where NOVA 4 was in the majority (56% versus 54%) and for Nutri-Score homogeneous distributions were found (between 16–27% in both studies). Regarding the cross-sectional results, percentages were not always similar; however, at the ordinal level, the distributions behaved similarly. As for other results, it is not possible to compare, since our work considered differences by food groups, showing the greatest inconsistencies between cereals and dairy products, which are precisely the products that are most altered by the biofortification implemented by the industry and proposed by health organizations to avoid malnutrition due to deficiencies [29]. Although the strategy has managed to solve some collective nutritional problems (hypovitaminosis), it should be planned with caution so as not to convert products that are not suitable for the population into alternatives to solve other problems [30]. Having detected the latter, the principal component analysis carried out in this study helps us to understand the dimensions targeted by each system according to food group. Interestingly, the theoretical definitions of each system can be identified by computer models, establishing that in most of the groups the warning labels is placed as an intermediary between NOVA and Nutri-Score. This proposes an interesting debate for all health authorities that are about to renew their dietary guidelines or dietary recommendations for their population, considering that they are aiming at dimensions, in some cases, totally contrary.

Regarding the methodological limitations of this research, the definition of the list evaluated could be an element of discussion; however, the nature of the construction of this list of foods constitutes the official body for the Ministry of Social Development and Family of the Government of Chile to determine the poverty line in the population. In this sense, it is aimed at compiling the minimum foods that every Chilean should be able to consume in order not to be vulnerable and is used for research on specific population groups [31]. In addition to the fact that foods were selected from all brands of a supermarket with wide national access, the final list of products evaluated constitutes an objective and representative selection of the supply to which most of the the country’s population is exposed. However, it may not represent the context of rural localities where food supply is limited to small grocery stores and not hypermarkets as in the country’s large cities. It would be interesting to assess such food contexts, and to verify whether they are exposed to a better or worse reality than cities.

As for the classifications made by each system, our work considered two qualification researchers for each one, to which a third actor was added to review the concordance of both classifications, exercising a weekly control point on the work carried out to rectify and unify criteria. In the case of FoP warning labels, it was quite simple, as it was necessary to corroborate the visual information provided by the front labelling with the nutritional composition declared by each product. In the case of Nutri-Score, a spreadsheet from the official body of the system (Sante Publique France^®^) [15] was used, ensuring that subjectivity or scoring errors by the researchers assigned to this task were minimized. NOVA is the most complex system to evaluate, since its parameters are not numerical, but rather purely qualitative. To overcome this problem, the expert judgement of one of the group’s researchers was used to verify each of the ratings made by the researchers in the field. In this area, the development of an algorithm to quantify the dimensions reported in this work would undoubtedly be a great tool to better advance research on ultra-processing and NOVA classification [32,33], from epidemiological to experimental studies.

Finally, regarding the concern about which system offers a better classification, the scope of our results does not allow us to answer this question, nor was it an objective of the present work. Our research was able to report concrete differences in the form and dimension of the classification of each system, providing critical input for authorities who must define which system they can implement in their future food and nutrition policies.

## 5. Conclusions

The present study was able to demonstrate that there are quantitative and qualitative differences between the classification and recommendation guidelines of the Nutri-Score, NOVA and FoP warning labels, finding concrete discrepancies between them. Differences can be explained by the orientation and objective of each classification, with stricter criteria being reported for the NOVA classification in the list evaluated in this research.

Our data do not allow us to establish which system has better information descriptors for consumers, but we can clarify that there are different degrees of sensitivity to classify in each of them. It is necessary to replicate research using a larger number of entries, as well as food listings different from those evaluated in the present work, as the results would possibly be different.

## Figures and Tables

**Figure 1 ijerph-19-14631-f001:**
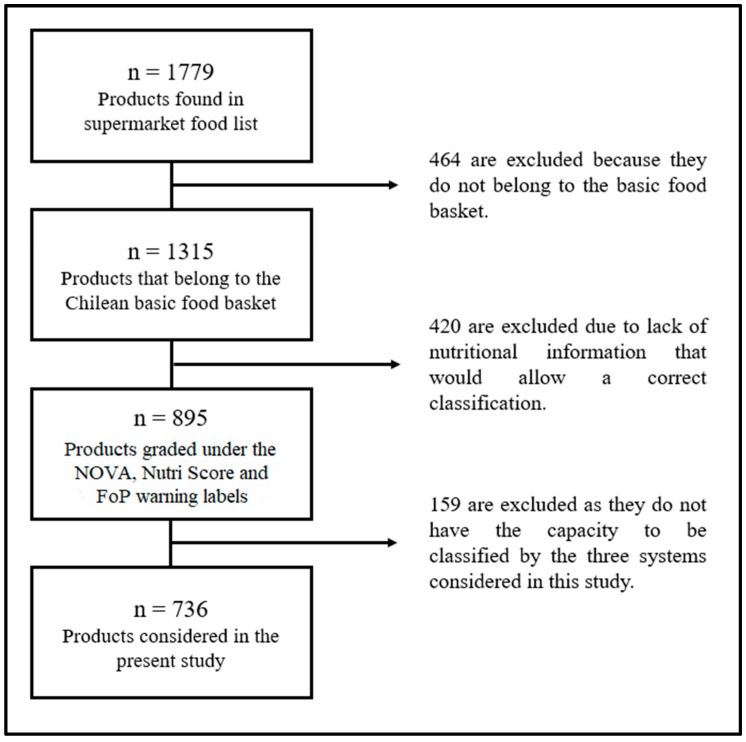
The workflow of food selection. The foods evaluated were selected from the online catalogue and included according to the inclusion criteria of belonging to the Chilean basic food basket regardless of the brand of the product, and excluded because they did not have complete nutritional information, were not available according to the establishment’s stock or could not be classified by the three systems evaluated (some bulk foods such as fruits, for example, for Nutri-Score guidelines).

**Figure 2 ijerph-19-14631-f002:**
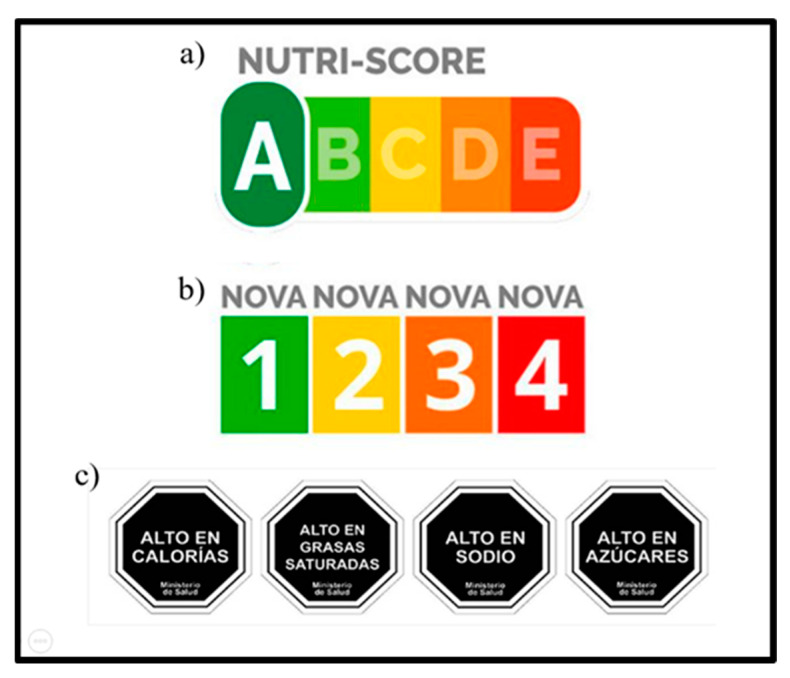
The food classification systems. This figure shows the classifications of each of the evaluated systems. First, (**a**) Nutri-Score categorises foods according to the balance of positive and negative nutritional characteristics for health, establishing the letters A and B as foods of high nutritional quality, and their counterparts D and E as products of low nutritional quality. For the (**b**) NOVA classification, four categories are established from number 1 to 4, basing its recommendation on the consumption of unprocessed foods (number 1) to processed foods (number 3), and suggesting the limitation or non-consumption of ultra-processed foods (number 4). Finally, (**c**) Chilean FoP warning labels establish their classification with cut-off points associated with critical nutrients that affect the health of the population, and precisely those foods that exceed the established limits receive a warning label that identifies them, and can be free of them or have all of them.

**Figure 3 ijerph-19-14631-f003:**
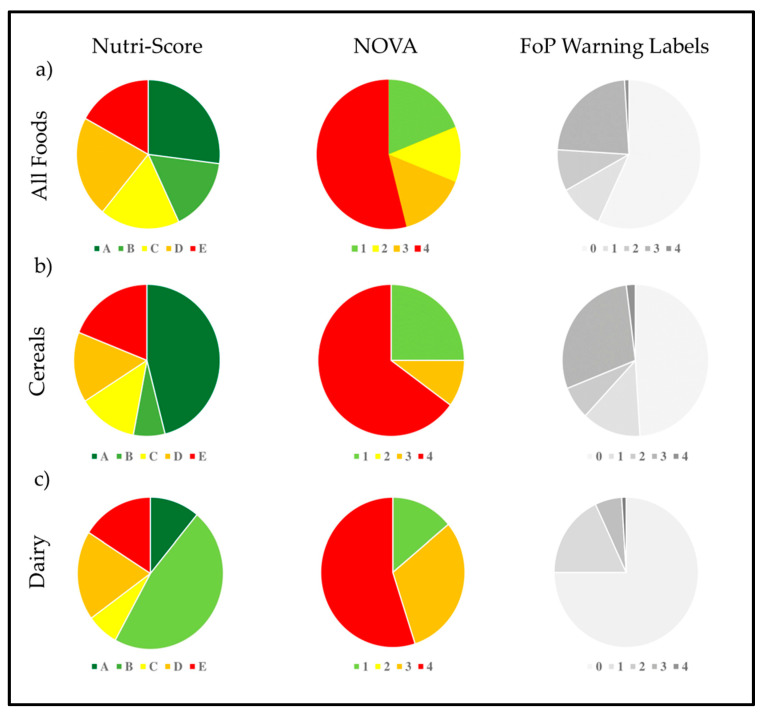
The distribution by different systems classification for all foods, cereals and dairy. All foods (**a**) are classified by Nutri-Score: A (27%), B (16%), C (18%), D (22%) and E (17%), for NOVA: 1 (19%), 2 (12%), 3 (15%) and 4 (54%), and FoP warning labels: 0 (57%), 1 (10%), 2 (9%), 3 (23%) and 4 (1%). For cereals (**b**), the Nutri-Score ranked: A (46%), B (7%), C (13%), D (15%) and E (19%), regarding NOVA show: 1 (25%), 2 (0%), 3 (10%) and 4 (65%), and FoP warning labels: 0 (49%), 1 (13%), 2 (7%), 3 (29%) and 4 (2%). For dairy products (**c**), NutriScore classified: A (11%), B (47%), C (7%), D (19%) y E (16%), for NOVA: 1 (14%), 2 (0%), 3 (31%) y 4 (55%), and for warning labels: 0 (75%), 1 (18%), 2 (6%), 3 (0%) and 4 (1%). Distribution by different systems classifications for protein foods, fats and sugars. Protein foods (meat, fish, eggs, and pulses) (**d**) are classified by Nutri-Score: A (60%), B (7%), C (6%), D (25%) and E (4%), for NOVA: 1 (48%), 2 (2%), 3 (25%) and 4 (25%), and for FoP warning labels as: 0 (71%), 1 (14%), 2 (6%), 3 (9%) and 4 (0%). Regarding Oils and Fats (**e**), ranked by Nutri.Score as: A (4%), B (5%), C (51%), D (36%) and E (4%), for NOVA: 1 (0%), 2 (64%), 3 (6%) and 4 (30%), and for FoP warning labels: 0 (50%), 1 (2%), 2 (21%), 3 (27%) and 4 (0%). Finally for Sugars and others (**f**), Nutri-Score stablished: A (4%), B (20%), C (18%), D (25%) and E (33%), for NOVA: 1 (6%), 2 (11%), 3 (18%) and 4 (75%), and for FoP warning labels: 0 (49%), 1 (3%), 2 (19%), 3 (38%) and 4 (0%).

**Figure 4 ijerph-19-14631-f004:**
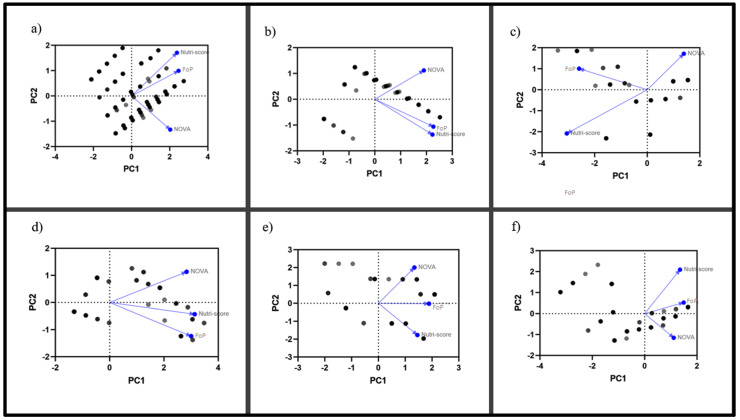
A Principal Component Analysis of each food system classification. In this figure, we proceeded to qualitatively analyse the systems assessed, identifying the dimensions targeted by a principal component analysis in (**a**) all foods, (**b**) cereals, (**c**) dairy, (**d**) protein foods, (**e**) lipids and fats, and (**f)** sugars and derivatives. This analysis establishes the ranking patterns used by NOVA, Nutri-Score and FoP warning labels, showing the closeness and remoteness between them. Black and grey boxes indicate the spatial location assigned for each input element, and blue arrows indicate the direction and dimension identified for each system.

**Table 1 ijerph-19-14631-t001:** A cross-ranking analysis between Nutri-Score and NOVA.

Nutri Score	NOVA 1	NOVA 2	NOVA 3	NOVA 4	Total
A	107 (54.3%)	3 (1.5%)	50 (25.3%)	37 (18.7%)	197
B	17 (14.4%)	3 (2.5%)	15 (12.7%)	83 (70.3%)	118
C	7 (5.4%)	39 (30.2%)	16 (12.4%)	67 (51.9%)	129
D	4 (2.4%)	40 (24.2%)	25 (15.1%)	96 (58.1%)	165
E	8 (6.2%)	4 (3.1%)	3 (2.3%)	112 (88.1%)	127
Total	143	89	109	395	736

In this table, a cross-analysis of the two classifications is carried out. In the rows, the foods categorised by Nutri-Score are shown, and those were distributed by each NOVA category, in order to establish concordance or discordance according to their recommendation parameters according to the nutritional quality established by their guidelines.

**Table 2 ijerph-19-14631-t002:** A cross-ranking analysis between Nutri-Score and Warning Stamps.

Nutri Score	FoP 0	FoP 1	FoP 2	FoP 3	FoP 4	Total
A	187 (94.9%)	8 (4.0%)	1 (0.5%)	1 (0.5%)	0 (0%)	197
B	101 (85.5%)	12 (10.1%)	4 (3.3%)	1 (0.8%)	0 (0%)	118
C	82 (63.5%)	21 (16.2%)	17 (13.1%)	9 (6.9%)	0 (0%)	129
D	39 (23.6%)	28 (16.9%)	36 (21.8%)	62 (37.5%)	0 (0%)	165
E	12 (9.4%)	4 (3.1%)	9 (7%)	98 (77.1%)	4 (3.1%)	127
Total	421	73	67	171	4	736

In this table, a cross-analysis of the two classifications is carried out. In the rows, the foods categorised by Nutri-Score are shown, and those were distributed by each FoP warning labels category, in order to establish concordance or discordance according to their recommendation parameters according to the nutritional quality established by their guidelines.

**Table 3 ijerph-19-14631-t003:** A cross-ranking analysis between NOVA and FoP Warning Labels.

NOVA	FoP 0	FoP1	FoP 2	FoP 3	FoP 4	Total
1	143 (100%)	0 (0%)	0 (0%)	0 (0%)	0 (0%)	143
2	69 (77.5%)	0 (0%)	12 (13.4%)	8 (8.9%)	0 (0%)	89
3	78 (71.5%)	18 (16.5%)	4 (3.6%)	9 (8.2%)	0 (0%)	109
4	131 (33.2%)	55 (13.9%)	51 (12.9%)	154 (38.9%)	4 (1.0%)	395
Total	421	73	67	171	4	736

In this table, a cross-analysis of the two classifications is carried out. In the rows, the foods categorised by NOVA are shown, and those were distributed by each FoP warning category, in order to establish concordance or discordance according to their recommendation parameters according to the nutritional quality established by their guidelines.

## Data Availability

A spreadsheet with all the products analyzed is available by sending an email to the corresponding author.

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
