# Peer review of "Discrepancy between Food Classification Systems: Evaluation of Nutri-Score, NOVA Classification and Chilean Front-of-Package Food Warning Labels"

_ijerph, 2022, doi:10.3390/ijerph192214631_

Round 1

Reviewer 1 Report

The article is new and interesting for those who are interested in the subject of food labeling and its relationship with public health policies.

Although it was carried out in Chile, to compare the stamp system (of Chile) with other forms of labelling, it is of international interest.

The manuscript needs to be improved in many aspects. Starting with English. Although I am not an English speaker, there are phrases and terms that need to be corrected. It also has flaws in the titles of tables and figures, The figures are blurred, unclear. I suggest using professional figure edits with higher resolution.

There is inconsistency in the use of naming the Chilean system. They use stamps, seals or “sellos” indistinctly. Lines 127, 243, 331, 349, 368 and others, plus figure 4: use the same way of naming the system, be it seals, stamp or SELLOS or warning stamps.

I detail my comments below:

Abstract

Line 20: may say “are still a matter of debate”

Lines 27-29: I understand that may be difficult to describe each category in the abstract, but a short description is recommended for understanding A and D. The same as you wrote for NOVA 1 and 4. A: better nutritional quality, D=Low nutritional quality.

Introduction

Line 42: The Americas is what exactly? Latin America and caribe? or all American continent?

Line 56: health bodies does not sound correct. Maybe use health authorities?

Line 57: was implemented by law (add by law, because you mention the law after, but ids not clear from where this is coming from)

Materials and methods:

Lines 88-93: Part of this paragraph should be in the description of the Fig 1. Everyone should be able to understand your figure, without reading the full text.

Figure 2: Describe better the figure. What is a), b) and c)? You may repeat in brief the description on the text. Everyone should understand the figure without reading the main text

Results:

Lines 159-162: This paragraph may be moved to the discussion. The results only describe the analyzed data. The next paragraph already describes the lack of concordance.

Figure 3: The figure is cut off. I suggest using a professional figure editor, with better resolution and that allows making a clearer and more compact panel. Please describe better the figure. What is a), b), c), d), e) and f)? You may also explain the labelling of each classification system. A,B,C,D E; 1,2,3,4.

Tables 1 and 2: Please, describe better the table. Meaning of NOVA 1,2,3,4, number of stamps and meaning of Nutri score A to E. Add "n" and (%) in each column. Improve the title, it must describe: what did you measure, variables shown, analysis done and sample used.

Figure 4: Same comment than previous figures. Describe better the meanings of the letters (a to f) and PC1-PC2. Sellos or Stamps? Black versus grey dots and blue dots?

Discussion:

Lines 254-267: The first paragraph does not correspond to a discussion section. I suggest reviewing it and move it to the introduction, avoiding repeating ideas. The discussion may start by resalting the results you found. And then, go deeper into them.

Line 271: no need to include (Figure 3). This was described in the results.

Line 282: recognized = acknowledged? Does not have the same meaning than in Spanish.

Line 344: When you mention representativeness, the sample may not be representative of rural population, for example, where no big supermarkets are available.

Reviewer 2 Report

The manuscript presents research on the qualitative and quantitative analysis of discrepancies between food classification systems: NOVA, Nutri-Score and Warning Stamps according to the Chilean basic food basket list. In my opinion, research in this area should continue, bearing in mind the increasing risk of diseases of civilization, especially in children and adolescents.  The paper will be suitable for acceptance after some improvement.

Below my comments to the authors:

1.      Results:

-        Figure 4 is not clearly described. It is not marked which point (a-f) corresponds to a specific group of analyzed products. One can only guess that the description is similar to Figure 3.

In addition, the comments on Figure 4 include the term Sellos system, which has not been previously introduced and described. It probably refers to the Seals system, but this should be corrected.

-        Page 9, line 245, 246: “Regarding cereals, this trend is repeated, and even a greater closeness between the Chilean system and the nutritional traffic light is exposed, as well as the trend that NOVA and Nutri-Score are further apart.”              The term "traffic light" in reference to the Nutri-Score system is unfortunate, as the NOVA system also relies on color signaling.

2.      Discussion

-        Page 10, line 256-258 “[95 % 256 CI, 6.8 % to 13.8 %] ; [95 % CI, 4.4 % to 10.9 257 %]; [95 % CI, 2.1 % to 8.3 %]. I do not understand the meaning of these expressions. Please give me an explanation or another notation.

3.      Conclusion

The authors do not settle the question of which system is better, because that was not the purpose of this work.  They made their assessment based on the realities of purchasing and availability of food products in Chile. If a similar study were conducted in another country, on another continent, would the results be similar? What could affect the dissimilarity of the results? What other research can be done on this topic, and what direction should it take?

4.      References

Please re-read the section References, some items need to be completed, e.g. there is no indication of pages.

Round 2

Reviewer 1 Report

The authors made all requested changes.

Congratulations! The quality improved.

The quality of the description of the figures allows a better understanding of them

Only two details remain. 1) line 63 add ( FoP) after Front-of-package and 2) line 236: it says results indicates, it should say results indicate (without the "s").

Author Response

Dear editor,

1) Added (FoP) on line 62.
2) Removed letter s in indicates (line 320).

Thank you very much.